

# A highly miniaturized satellite payload based on a spatial heterodyne spectrometer for atmospheric temperature measurements in the mesosphere and lower thermosphere.

Martin Kaufmann[1,2], Friedhelm Olschewski[2], Klaus Mantel[3], Brian Solheim[4], Gordon Shepherd[4], Michael Deiml[1,2,a], Jilin Liu[1,2], Rui Song[1,2], Qiuyu Chen[2], Oliver Wroblowski[1,2], Daikang Wei[1,2], Yajun Zhu[1], Friedrich Wagner[8], Florian Loosen[3,8], Denis Froehlich[5], Tom Neubert[5], Heinz Rongen[5], Peter Knieling[2], Panos Toumpas[2], Jinjun Shan[6], Geshi Tang[7], Ralf Koppmann[2], and Martin Riese[1]

[1]Institute of Energy and Climate Research (IEK-7), Research Center Juelich, Juelich, Germany

[2]Institute for Atmospheric and Environmental Research, University of Wuppertal, Germany

[3]Max Planck Institute for the Science of Light, Erlangen, Germany

[4]Centre for Research in Earth and Space Science, York University, 4700 Keele Street, Toronto, Ontario, Canada

[a]now at: OHB System AG, Bremen, Germany

[5]Central Institute for Engineering, Electronics and Analytics, Electronic Systems (ZEA-2), Research Centre Juelich, Juelich, Germany

[6]Department of Earth and Space Science and Engineering, York University, Toronto, Ontario, Canada

[7]Aerospace Flight Dynamics Laboratory, Beijing, China

[8]Institute of Optics, Information and Photonics, Friedrich-Alexander-Universitaet Erlangen-Nuernberg, Erlangen, Germany

**Correspondence:** Martin Kaufmann (m.kaufmann@fz-juelich.de)

**Abstract.** A highly miniaturized limb sounder for the observation of the $O_2$ A-Band to derive temperatures in the mesosphere and lower thermosphere is presented. The instrument consists of a monolithic spatial heterodyne spectrometer (SHS), which is able to resolve the rotational structure of the R-branch of that band. The relative intensities of the emission lines follow a Boltzmann distribution and the ratio of the lines can be used to derive the kinetic temperature. The SHS operates at a Littrow wavelength of 761.8 nm and heterodynes a wavelength regime between 761.9 nm and 765.3 nm with a resolving power of about 8,000 considering apodization effects. The size of the SHS is 38x38x27 mm$^3$ and its acceptance angle is $\pm 5^o$. It has an etendue of 0.014 cm$^2$ sr. Complemented by a front optics with a solid angle of $0.65^o$ and a detector optics, the entire optical system fits into a volume of about 1.5 liters. This allows to fly this instrument on a 3 or 6 unit CubeSat. The vertical field of



view of the instrument is about 60 km at the Earth's limb if operated in a typical low Earth orbit. Integration times to obtain an entire altitude profile of nighttime temperatures are in the order of one minute for a vertical resolution of 1.5 km and a random noise level of 1.5 K. Daytime integration times are one order of magnitude shorter. This work presents the design parameters of the optics and a radiometric assessment of the instrument. Furthermore it gives an overview of the required characterization

and calibration steps. This includes the characterization of image distortions in the different parts of the optics, flat fielding and the spectral power estimation.

# 1 Introduction

Atmospheric waves drive important atmospheric circulation patterns such as the Brewer-Dobson circulation in the stratosphere

and mesosphere. Wave structures are detectable in atmospheric wind and temperature fields. Small-scale gravity waves are particularly important in the mesosphere and even lower thermosphere.

To demonstrate new ways to measure atmospheric waves at high spatial resolution, Song et al. (2017) presented a new satellite observation strategy for the detection of gravity waves in the mesosphere and lower thermosphere (MLT). This measurement mode requires an agile satellite platform to make multi-angle observations of a particular atmospheric volume and a

spectrometer particularly suited for the detection of faint emission lines.

The concept and optical layout for such an instrument is presented, which fits onto a nano-satellite platform, such as a CubeSat (e.g., Poghosyan and Golkar, 2017, and references therein). To customize an instrument to the constraints of a CubeSat gives access to a variety of standardized satellite-bus components and flight opportunities, because CubeSat deployers are nowadays an integral part of many launch vehicles. In return for these advantages, the payload has to cope with very restricted

mass, volume, and power resources.

The most common technique to obtain temperatures in the upper mesosphere and lower thermosphere is to measure the emission of $CO_2$ in the mid infrared or to measure the absorption of sunlight by $CO_2$. Although the modeling of $CO_2$ emissions has its own problems regarding the determination of the non-local thermodynamic equilibrium state of $CO_2$, this method is well accepted and gives temperatures over a broad altitude range at a good signal to noise ratio. The most prominent instruments

using infrared emissions to derive MLT temperature are ISAMS (Nightingale and Crawford, 1991), CRISTA (Offermann et al., 1999; Grossmann et al., 2002), MIPAS (Fischer et al., 2008), and SABER (Russell et al., 1999) for emission measurements and HALOE (Russell et al., 1994) and ACE-FTS (Bernath, 2017) for occultation measurements.

Instruments measuring at infrared or longer wavelengths are quite large or high energy consuming, so that measurements in the ultraviolet/visible/near-infrared spectral regime are most appropriate for a CubeSat platform. In this wavelength regime,

mesospheric temperature measurements can be performed by the evaluation of the rotational distribution of a molecular emission band. The emitting states should be sufficiently long-lived, and the rotational distribution should be thermalized, such that



it can be described by the kinetic temperature. It is best, if this emission is visible during day- and nighttime, such that temperatures can be obtained at all local times. The $O_2$ atmospheric band system fulfills all of these requirements. The strongest band within this system is the the $O_2$ (0,0) atmospheric A-band at 762 nm, which was investigated in several studies (e.g., Rodrigo et al., 1985; Torr et al., 1985; McDade and Llewellyn, 1986; Meriwether, 1989; Slanger and Copeland, 2003). The $O_2$ A-band

has been used to derive global MLT temperatures in recent years using HRDI/UARS Fabry-Perot interferometer data (Ortland et al., 1998) and OSIRIS/ODIN grating spectrometer data (Sheese et al., 2010).

This temperature measurement technique builds upon relative intensity measurements. The requirements to monitor the radiometric performance of such kind of instrument are much more relaxed than for measurement strategies which rely on absolute intensities. Another advantage is that the A-band emits at wavelengths below 1 $\mu$m, so that silicon-based detectors

operating at ambient or moderately cooled conditions can be used for detection. This reduces the power consumption, mass, and costs of such an instrument significantly.

In this work we give an overview of the design of a highly miniaturized instrument to measure $O_2$ A-band limb radiances. We summarize various topics on the radiometric and optical design as well as the calibration and processing of the data. Further and more detailed studies on these subjects are currently in preparation for publication. We are preparing such an instrument

for a detailed laboratory characterization and an in-orbit verification in the near future. Beside putting the individual works on this instrument into a broader context, the purpose of this paper is to put the design and analysis methods, which are partly different from previous work, up for discussion in the scientific community.

## 2  $O_2$ Atmospheric Band Emissions

Light emitted in the $O_2$ atmospheric band system stems from the transition of $O_2(b^1\Sigma_g^+)$ to $O_2(X^3\Sigma_g^-)$. There are three

absorption bands in this system (A, B, and $\gamma$ bands). All of these bands end up in a vibrational ground state. The upper states are at v=0, 1, 2 for the A, B, and $\gamma$ bands, respectively. None of these bands can be observed from the ground because of the high abundance of ground state molecular oxygen molecules in the atmosphere. The radiative lifetime of the $O_2(b^1\Sigma_g^+)$ state is about 12 seconds (Burch and Gryvnak, 1969). This long lifetime assures that the molecule is in rotational equilibrium with the ambient atmosphere, such that rotational and ambient temperature are identical. An overview of the chemistry and

molecular dynamics of excited $O_2$ is given by, e.g., Slanger and Copeland (2003) and references cited therein. It can be briefly summarized as follows: $O_2(b^1\Sigma_g^+)$ is excited by collisions of ground state $O_2$ with $O(^1D)$, which is produced in the photolysis of $O_2$ in the Schumann-Runge Continuum and in the photolysis of $O_3$ in the Hartley band. Due to the long radiative lifetime of $O(^1D)$ (about 2 minutes), most of the energy of $O(^1D)$ is lost by quenching with $N_2$ and $O_2$, producing a multitude of excited $N_2$ and $O_2$ states, including the ones emitting the atmospheric band system. Another excitation mechanism of $O_2(b^1\Sigma_g^+)$ is

resonance scattering or absorption of photons in the atmospheric bands itself. The third process is the collision of ground state $O_2$ with a metastable, highly excited state of $O_2$ produced in the recombination of two atomic oxygen atoms. This two step process was first proposed by Barth and Hildebrandt (1961); Barth (1964). It is the only excitation process which is active during day- and nighttime. Figure 1 shows simulated volume emission rates of $O_2(b^1\Sigma_g^+)$ separated by excitation processes,





as simulated with the model described by Song et al. (2017). According to these simulations, the day- to nighttime ratio of the $O_2(b^1\Sigma_g^+)$ number densities is about a factor of 50 in the vicinity of the mesopause.

The spectral shape of the A-band for two different temperatures is illustrated in Figure 2. Higher temperatures give a flatter spectrum. A 10 K change in temperature affects the rotational distribution of strong emission lines at 760–765 nm between 5  $\pm6\%$. This means that the band structure must be measured better than 1% to derive temperatures with a precision of 1.5 K.

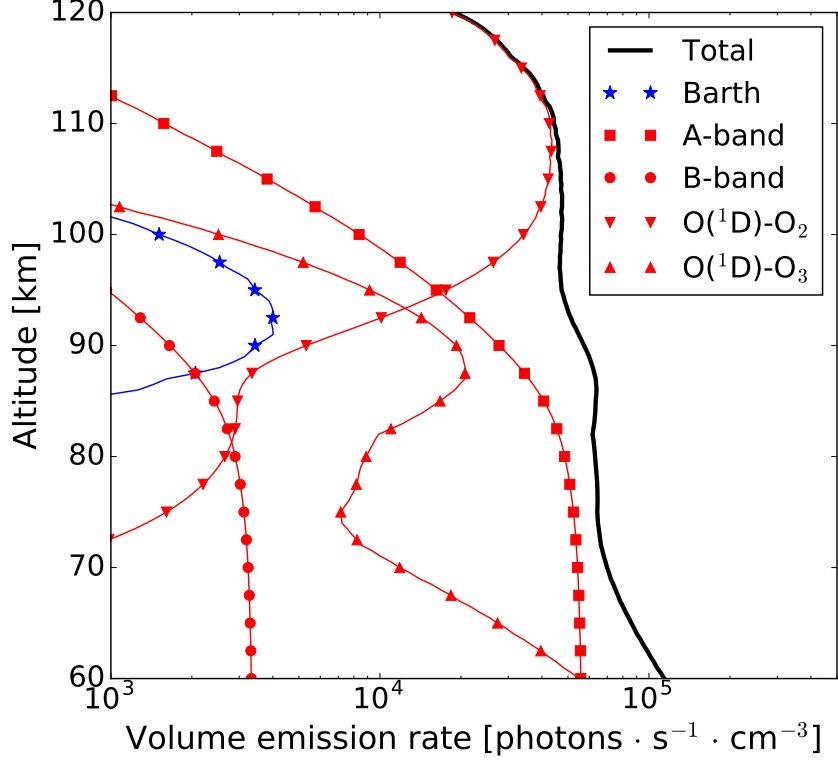

**Figure 1.** Number density of $O_2(b^1\Sigma_g^+)$ separated by excitation processes. 'Barth' indicates the number of molecules created by the recombination of atomic oxygen. This is the only excitation process being active during day and night. 'A-Band' and 'B-Band' label the fraction of molecules excited by resonance absorption in those bands. 'O$_2$' and 'O$_3$' mark the excitation by collisions with O($^1$D), which is created by photolysis of $O_2$ and $O_3$, respectively

## 3  Spatial Heterodyne Spectrometer

At the beginning of this project, different instrument concepts were considered to detect the mesospheric A-band limb emissions from a CubeSat (Deiml et al., 2014). For a variety of reasons, it was decided to realize the instrument with a spectrometer. Performance considerations lead to the selection of a Fourier transform spectrometer (FTS) . With its compact and monolithic





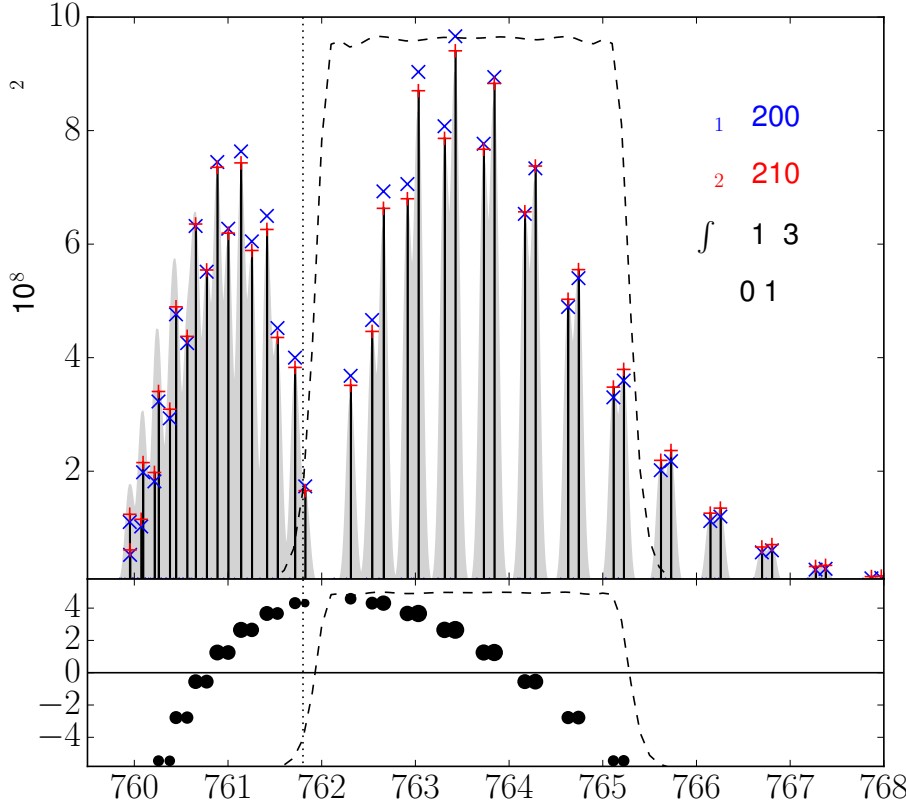

**Figure 2.** $O_2$ A-band limb emission line calculations assuming a global temperature of 200 K and 210 K (upper panel). The spectra in the upper panel have been normalized to show identical band intensities. The vertical bars and the blue plus signs mark the emission line intensities for 200 K, the light gray area shows their intensities as seen from an instrument with a spectral resolution of 0.1 nm (multiplied by a factor of 10). The red x signs show line intensities for 210 K. The dashed line is the filter transmission curve of the instrument presented later. The dotted vertical line is drawn at the Littrow wavelength. Within the filter, more than 50% of the total band intensity (at 200 K) are emitted (97 out of 183 photons/s/cm$^2$/sr). The percentage difference of the line intensities at 200 K and 210 K are shown in the lower panel; the symbol size scales with the absolute intensity of the lines. The atmospheric background data is taken from the HAMMONIA model, and the spectroscopic data stems from the HITRAN database (Gordon et al., 2017).

design, a spatial heterodyne spectrometer (SHS) deemed the most appropriate candidate, in accordance with the findings of Watchorn et al. (2014) in the framework of another study.

In principle a SHS is a Fourier transform spectrometer, where the mirrors in each arm are replaced by diffraction gratings (Figure 3). The incoming wavefront is diffracted at the gratings, with a wavelength-dependent angle. The superposition of the





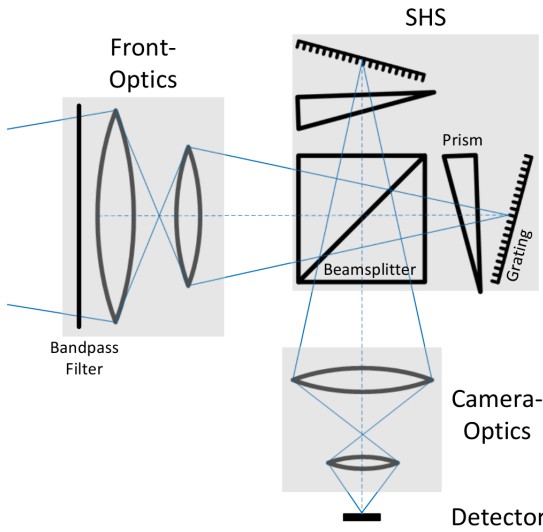

**Figure 3.** Principle design of the SHS with front and detector optics.

two wavefronts then produces straight, parallel, and equidistant fringes with a spatial frequency depending on the wavelength of the light. The zero frequency of the fringe pattern is at the Littrow wavelength and the spectral peaks of the neighboring wavelengths are spread or heterodyned around this central wavelength. The concept was originally proposed by Pierre Connes in a configuration called "Spectromètre interférential à selection par l'amplitude de modulation (SISAM)" (Connes, 1958). With the advent of imaging detectors, this idea was taken up by, e.g., Harlander and Roesler (1990); Douglas (1997); Smith and Harlander (1999); Watchorn et al. (2001); Harris et al. (2004); Roesler (2007); Watchorn et al. (2010); Bourassa et al. (2016); Lenzner and Diels (2016). The design of a SHS for a particular wavelength and spectral resolution follows a few simple relations, which are shortly summarized to illustrate the main characteristics of this device. For a derivation of the mathematical expressions see, e.g., Harlander (1991); Cooke et al. (1999); Smith and Harlander (1999), and references cited therein:

The tilt angle of the gratings with respect to the optical axis is called Littrow angle $\Theta_L$. Light at the Littrow wavenumber $\sigma_L$ is returned in the same direction as the incoming path, as described by the grating equation (for diffraction order one):

$$\sigma_L = \frac{1}{2\sin\Theta_L} \tag{1}$$

Combining the intensity equation of a conventional FTS and the grating equation for small incident angles at the grating gives the SHS equation for ideal conditions, relating the incoming radiation $S$ at wavenumber $\sigma$ to the spectral density $I$ at



position $x$:

$$I(x) = \frac{1}{2} \int_\sigma S(\sigma)[1 + \cos 2\pi\,(\kappa x)]\, d\sigma \qquad (2)$$

$\kappa$ is the heterodyned fringe frequency:

$$\kappa = 4 \tan \Theta_L (\sigma - \sigma_L) \qquad (3)$$

The maximum resolving power $R$ of an SHS is nearly proportional to the number of grating grooves illuminated by the incoming beam or in other words the illuminated spot size $W$ on the grating multiplied by the grating groove density $g$ times two:

$$R = 2Wg \qquad (4)$$

In practice, the effective spectral resolution or bandpass is often limited by the detector resolution. The maximum fringe
frequency shall not exceed half the pixel frequency of the detector. This means, that the spectral range $\lambda_{max} - \lambda_{min}$, which can be detected for given spectral resolution $\Delta\lambda$, has to be lower than half the pixel number $N$:

$$\frac{\lambda_{max} - \lambda_{min}}{\Delta\lambda} \leq \frac{N}{2} \qquad (5)$$

As for conventional FTS or Fabry Perot instruments, the acceptance angle of light for a conventional SHS is inversely proportional to its resolving power $R$ (e.g., Harlander, 1991), which is a few orders of magnitude larger than for conventional
grating spectrometers of the same size. The acceptance angle of an SHS can be increased significantly, if prisms are inserted into the two interferometer arms. This configuration was first implemented for upper atmospheric temperature measurements by Hilliard and Shepherd (1966) with a Michelson interferometer. The prisms rotate the image of the gratings so that they appear to be located in a common virtual plane which is oriented perpendicular to the optical axis for a wide range of incident angles. At the end, the acceptance angle of the SHS including field widening prisms is only limited by spherical aberration for
systems with small Littrow angles and astigmatism for large Littrow angles (Harlander et al., 1992). Depending on the actual design, the prisms increase the etendue or throughput of an SHS by 1–2 orders of magnitude. The calculation of the prism apex angle is given by, e.g., Harlander et al. (1992).

A general advantage of SHS are the relaxed alignment tolerances, because in most optical setups the gratings are imaged onto a focal plane array. As a result, each detector pixel sees only a small area of the optical elements, so that moderate misalignments
or inaccuracies in the surface quality affect limited spatial regions on the detector, only. This means that the interferogram is distorted locally rather than reduced in contrast. The main benefit of the SHS is that they can be built monolithically, making them very robust for harsh environments, e.g. during rocket launches.

The basic design parameters of the SHS were calculated analytically using the SHS equations mentioned above. The materials of the optical glass components, the apex angle of the prisms as well as the distances between the various components



| attribute | property |
|---|---|
| **fore optics** (incl. filter) | |
| wavelength range | 761.9–765.3 nm |
| clear aperture | $\pi \, (33 \text{ mm})^2$ |
| field of view | $\pm 0.65^o$ |
| etendue | $0.014 \text{ cm}^2 \text{ sr}$ |
| focal length | 136 mm |
| image size | $\pi \, (3.5 \text{ mm})^2$ |
| **SHS** | |
| grating groove density | 1200 lines/mm |
| Littrow wavelength | 761.8 nm |
| Littrow angle | $27.2^o$ |
| field of view | $\pm 5^o$ |
| **detector optics** | |
| numerical aperture (obj. space) | 0.12 |
| magnification | 0.55 |
| focal length | 28 mm |
| length of imaging system (incl. SHS) | 75 mm |
| **detector** | |
| total pixel count | 1920 x 1080 |
| used pixel count | 840 x 840 |
| pixel size | $5.04 \times 5.04 \; \mu\text{m}^2$ |
| quantum efficiency | 0.4 at 760 nm |
| dark current per pixel at $20^o$C | 2-4 $e^-$/s |
| readout noise (rms) | 1 $e^-$ |
| **performance** | |
| optical resolving power | 16,800 |
| expected resolving power (approx.) | 8000 |

**Table 1.** Summary of optics and filter properties

were optimized and iterated by means of optical ray tracing software (ZEMAX). The resulting basic design parameters are summarized in Table 1. A simulated interferogram of the $O_2$ A-Band as seen from this instrument is illustrated in Figure 4.

An integral part of an SHS design is the optical filter located between the SHS and the scene to be observed. For this instrument, a six cavity design bandpass filter with a center wavelength of 763.6 nm and a bandwidth of 3.3 nm was chosen.

5 The filter is illuminated at an angle of incidence of $\pm 0.65^o$, resulting in a blue shift of 0.8 nm. The temperature coefficient of





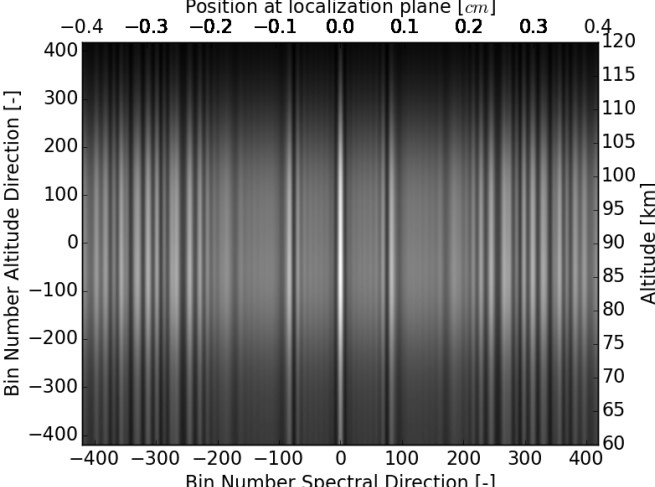

**Figure 4.** Simulated interferogram of the $O_2$ A-Band emission for various altitudes ((Note: I will update the figure later on))

this filter is 5 pm/K, resulting in a spectral shift of the bandpass of 0.3 nm between -10$^o$ and +50$^o$. Since an SHS instrument maps the spectrum on both sides of the Littrow wavelength symmetrically into Fourier space, the filter must be adapted in such a way that there is no overlap of lines from different sides of the Littrow wavelength in the interferogram. In our design, the Littrow wavelength is at 761.8 nm, e.g. the filter blocks most of the radiance from the shorter wavelength side of the Littrow wavelength (Figure 2).

The SHS design was performed in a collimated configuration, because that is most easy to understand and to simulate, although the device is operated in a focused configuration later on. The basic design parameters are not affected by this assumption, but the performance is, because spherical waves, as they pass through an SHS in a focused design, suffer more aberration than parallel waves.

## 4   Front and Detector Optics

The purpose of the front optics is to image a scene at the Earth's limb onto the gratings. The detector optics images the gratings onto the focal plane of the 2-dimensional detector. The image at the detector contains spatial information about the scene in both dimensions. An interferogram is superimposed on this scene in the direction perpendicular to the grating grooves. For the instrument presented in this work, the gratings are oriented in such a way that the interferogram spans over the horizontal direction, assuming that intensity fluctuations in the horizontal direction are small compared to the modulation depth of the interferogram, which is valid in atmospheric limb sounding. The front-optics (Figure 5) consists of four lenses, which image an object at infinity of an angular extent of 1.3$^o$ onto a circle of a diameter of 7 mm on the virtual image of the gratings. This corresponds to a theoretical spectral resolution of about 16,800 (Equation 4). The clear aperture of the front lens is 66 mm and the distance between the first lens and the SHS is 104 mm. The etendue of this configuration is 0.014 cm$^2$ sr.





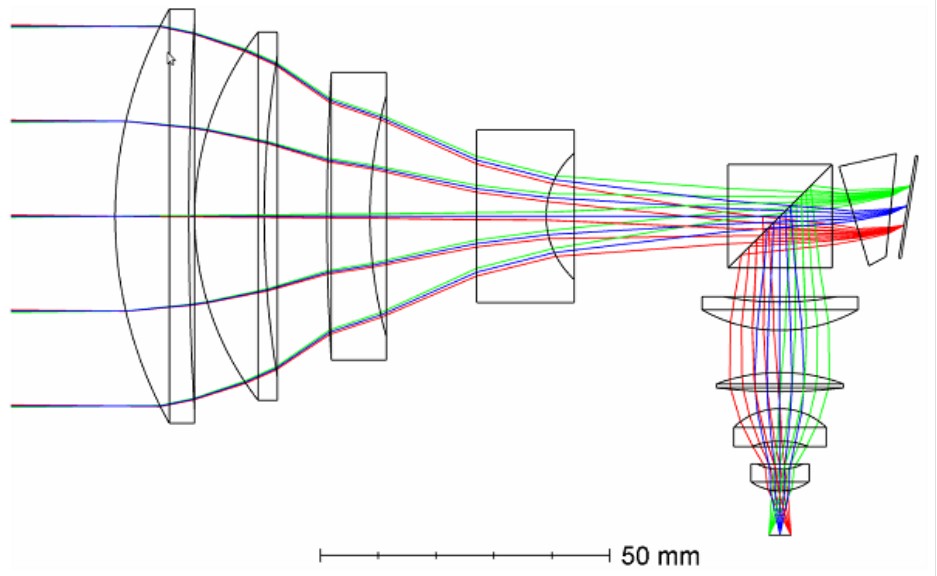

**Figure 5.** Optical components of the instrument, including the interference filter, the front optics, the SHS, the detector optics and the detector. The interference filter (not shown) is located in front of the leftmost lens. Only one arm of the SHS is shown.

The detector optics images the active area of the gratings onto the detector and consists of lenses as well. The magnification is 0.55, i.e. the illuminated area at the detector has a diameter of about 3.8 mm. This value was chosen as a trade-off between the form factor required and the desired spectral and spatial resolution (see below). The distance between the beam splitter and the detector focal plane is 46 mm.

The aperture stop of the optical system, which limits the amount of light passing through the instrument, is the mounting of the first lens of the front optics. In the current version of the instrument, there is a Lyot stop after the last lens of the detector optics.

The detector chosen for this instrument is a low noise silicon-based CMOS image sensor from Fairchild Imaging (HWK1910A). The optical format is 2/3 (9.7 mm x 5.4 mm) and the pixel size is 5 $\mu$m x 5 $\mu$m, resulting in 1920 x 1080 pixels in total, from those 840 x 840 pixels are used here. The quantum efficiency of this detector is about 0.4 at 760 nm.

Like the SHS, the entire optical system was optimized using optical raytracing software as well. The wavefront peak-to-valley extension of the optical system is less than a half wavelength for center rays and one wavelength at maximum for the edge region of the field. The extension of the point spread function is 5 $\mu$m for inner and 10 $\mu$m for outer pixels, which does not deteriorate the determination of the different waves in the interferogram, because the highest spatial frequency to be observed has a wavelength of about 70 $\mu$m. Optical distortions introduced in the common optical path are not part of the optimization procedure, because they can be removed in the post-processing or calibration of the instrument.

To evaluate the spectrometric performance of the system, the differences in the phase distortion of the two arms are most relevant, because this would result in an irreversible loss of contrast. This quantity is not directly accessible by the figures of



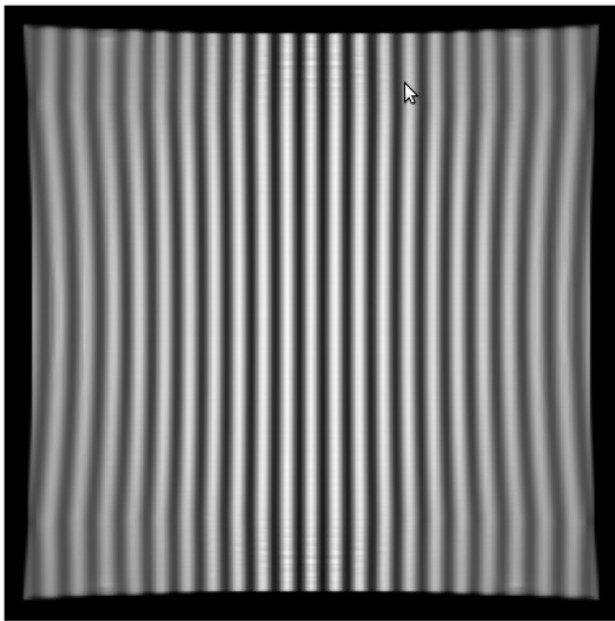

**Figure 6.** Simulated interferogram for a focused configuration considering wave aberrations

merit mentioned above. This effect was investigated by calculating the wave aberrations in the exit pupil of the system for object and reference arm, respectively. The corresponding complex amplitudes are then superposed and propagated into the detector plane by a Fourier transformation. Taking the absolute square of the resulting amplitude gives the intensity distribution for the corresponding light source point (Figure 6). The simulation indicates, that the contrast is reducing towards the edges of

5    the field, as expected. A comparison of these simulations and real measurements is given in Mantel (2017).

## 5    Performance Assessment

To determine the expected signal-to-noise ratio of the instrument for a given integration time, we estimate the amount of incoming light, which is available in the modulated part of the interferogram and the noise of the detector. In a SHS, 50% of the incoming radiation are lost at the beam splitter. The holographic gratings used have an efficiency of about 2/3 at 765 nm, so

10    that another 1/3 of the radiation is not available in the modulated part of the radiance. Misalignments and aberrations of optical components are estimated to reduce the contrast of the interferogram, so that we expect to detect about 20% in the modulated part of the interferogram.

The limb radiances of a strong line in the $O_2$ A-band nightglow maximum recorded over an altitude range of 1.5 km are about $1 \cdot 10^9$ photons/(s cm$^2$ sr) (c.f. Figure 2). Considering the etendue of the system and the fraction of the detector illuminated by

15    this emission layer (about 2%), and assuming that 20% of the photons end up in the modulated part of the signal, this yields to about 80 photons/s at every spectral point within the interferogram. Since the intensities from a vertical layer of 1.5 km





thickness illuminate 20 detector rows, the average signal introduced by one emission line on an individual detector pixel is 4 photons/s. Considering that the detector records light from all spectral elements within the bandpass of the instrument, and some radiance will end up in the unmodulated part of the interferogram, each detector pixel will record 10–20 times these values in total.

The noise of the signal is, by far, limited by shot noise, which scales with the square root of the (electrical) signal. The latter consists of the electrons excited by the signal of interest and the dark current caused by thermal processes. According to our own measurements, the dark current of the detector is 2–4 e$^-$/s/pixel (corresponding to a photon flux of 5-10 photons/s/pixel) at 20°C, which is a factor of 7–10 lower than the threshold given by the manufacturer. The dark current reduces a factor of two every 7 K (Liu et al., 2017). Although the dark current at a detector temperature of 20°C is a factor of 2.4 larger than

the expected atmospheric signal in the nightglow maximum, it does not deteriorate the data processing significantly. At this temperature, it is also not a dominant source of random noise, because the effect of the noise introduced by multiplexing is a factor of 5–10 larger. This changes with higher temperature. Therefore the detector should be operated below 20°C. The readout noise of the detector was measured to 1 e$^-$, which is in agreement with the specification given by the manufacturer. Taking into account, that 20 detector rows at summed up for each altitude bin, this noise component is negligible for integration

times larger than one second compared to the shot noise.

Considering the intensity of the A-band signal of the nightglow layer maximum and the detector performance, the expected signal-to-noise ratio for vertical resolution of 1.5 km and an integration time of 60 s will be between 12 and 24, depending on the actual performance of the instrument. A signal-to-noise-ratio of slightly less than 10 is required to reconstruct the spectral power at a precision of 1% (c.f. Chapter 7), which is needed to derive temperature with a noise error of 1.5 K (Song, personal

communication, 2017).

## 6   Instrument Characterization

The conversion of the detector signal into calibrated spectra involves a number of calibration steps. As pointed out in the previous section, the modulated to unmodulated signal ratio is one of the key points here. To quantify this ratio, the SHS equation for idealized conditions (Equation 2) has to be extended (Englert and Harlander, 2006):

$$I = I_{\text{modulated}} + I_{\text{non-modulated}} \tag{6}$$
$$I_{\text{non-modulated}} = \int_0^\infty S(\kappa)R(\kappa)\left[t_A^2(x) + t_B^2(x)\right]d\kappa$$
$$I_{\text{modulated}} = \int_0^\infty 2S(\kappa)R(\kappa)\epsilon(x,\kappa)t_A(x)t_B(x)\cos\left[2\pi\kappa x + \Delta(x,\kappa)\right]d\kappa$$

For better clarity, the integration variable in these expression is the heterodyned fringe frequency $\kappa$ instead of wavenumber $\sigma$, which is a normal linear dependency. One of the extensions compared to Equation 2 is the introduction of different intensity





transmission functions $t_A$ and $t_B$ for the two SHS arms. In addition, a term $\epsilon(x,\kappa)$ was added, which considers that the modulation efficiency can depend on the location within the interferogram and its frequency. Finally, a phase distortion term $\Delta(x,\kappa)$ quantifying any phase and frequency distortions within the interferogram was introduced. The characterization of some of these quantities is closely linked to the imaging capabilities of the system, because they have the same origin, namely

imperfections or misalignments of the optical system.

The following sections give an overview about the calibration steps to be performed and the algorithms to describe and correct for some of the instrument effects in view of the generation of calibrated spectra.

### 6.1    Flatfielding

The first step in the calibration sequence is flatfielding. In our terminology, the flat-field defines the radiometric uniformity

of the system or parts of it without the occurrence of interference. This corresponds to the 'non-modulated' intensity term. Non-uniformity can be caused by different sensitivities of detector elements, or inhomogeneities within optical components. Other reasons can be any kind of misalignment of the optical components including the SHS. Flatfielding is essential for the interferometric as well as the spatial analysis of the data.

Englert and Harlander (2006) give an overview about different flatfielding approaches. The first step in our flatfielding

procedure is to expose the detector to a uniform radiation source to identify hot and dead pixels, and to adjust the offset and gain values of each pixel in such a way that a uniform image is obtained. In a second step called 'balanced arm flatfielding approach' (Englert and Harlander, 2006), the entire instrument is illuminated with a uniform radiation source and one by one SHS arm is blocked.

### 6.2    Image and phase distortion correction

Large scale image distortions such as bending affect the distribution of spatial information and modify the frequency of the interferogram at the same time. Aberrations can also alter the foci of the image elements, affecting the local contrast of the interferogram, thus changing the ratio of the modulated and unmodulated intensities across the field. This affects the spectral power in each frequency as well as the overall spectral resolution of the system.

Due to the highly compact design of the detector optics and the use of spherical lenses only, significant image distortions

are expected. To characterize image distortions of the entire optical system a line grid target will be positioned in front of the instrument. Then, the SHS arms are blocked one by one to record two images of the test target. The division model (Fitzgibbon, 2001) will be used to correct for the spherical symmetric distortions. Within this model, radial distortion coefficients are fitted to straighten lines in the image. These measurements will also verify the geometrical point spread functions, which are expected to be much smaller than the required spatial resolution of the instrument. A computer analysis predicts that the image distortions

introduced by the front optics are less than 10% than those introduced by the detector optics.

Another measurement, which will be used to quantify the image distortion of the SHS and the detector optics, are the interference fringes introduced by a monochromatic light source. In the general case, the bending of those fringes is frequency dependent. From an interferometric point of view, those distortions introduce a phase- and frequency shift of the interferogram




along a detector row. Englert et al. (2004) present a method to compensate for this effect by measuring a reference spectrum, which is used to derive a correction function. The uncorrected interferograms are Fourier-transformed and convolved with this function. Since we have a tunable laser source available for the entire spectral range of the instrument, we use another approach. The laser emission is coupled into a large integrating sphere, which will provide a homogeneous flat field of the laser light. The laser frequency and power is continuously monitored during the measurement. The laser power and the flux at the exit of the sphere are calibrated before the measurements are taken. These measurements allow to quantify and correct for various effects: First, they can be used to verify the flatfielding done before by blocking the SHS arms one by one. Second, these measurements can be used to correct the image distortions of the SHS and the detector optics. The reference image is generated from an interferogram by defining a certain intensity level as an edge. The edges correspond in a sense to the reference lines of a test image, which can be used to derive a distortion correction. Another approach is to fit a linear (or higher polynomial order) correction term to each detector row. Both methods can consider a frequency dependency of the phase or of an image distortion by tuning the wavelength of the laser. The third purpose of these measurements is to quantify the contrast or visibility $\nu$ of the interferograms (Shepherd, 2002), which is defined as the amplitude of the modulation normalized to the average signal (times two):

$$\nu = \frac{I_{\max} - I_{\min}}{I_{\max} + I_{\min}} \tag{7}$$

The visibility depends on several factors, such as internal straylight, the grating performance, surface or material properties or imperfections, misalignments, contamination, etc. Visibility depends on the modulation transfer function and can be frequency-dependent. It can also vary across the field as a consequence of strong aberrations or misalignments of the system. Since the total power in each wave is needed for temperature retrieval, the visibility calibration is as important as a radiometric calibration, and it can even cover the radiometric calibration, if the power of the laser scene is known good enough.

To get the modulation or envelope function of the monochromatic interferogram as needed for the visibility calibration, we calculate its Hilbert-transform. The sum of the signal and its Hilbert-transform as imaginary part gives an analytic or holomorphic representation of the interferogram (e.g., Feldman, 2011). The absolute value of this complex-valued signal gives the instantaneous amplitude or envelope of the signal. More details on the application of this method are given in Liu et al. (2017).

## 7 Spectral Power Estimation

For operating the instrument a compromise between the (horizontal) sampling frequency and the signal-to-noise ratio (SNR) of individual measurements has to be made. Therefore, key criteria for the selection of suitable spectral density estimators are a good performance for noisy data and the conservation of the spectral power. The most obvious method to convert the measured interferogram into a spectrum is Fourier transformation. But its efficiency for noisy signals is lower than it is for more advanced methods. Since the number of frequencies in the interferogram is limited, known a-priori and can be verified from time to time, eigenspace methods are promising candidates. They split the autocorrelation matrix into orthogonal signal





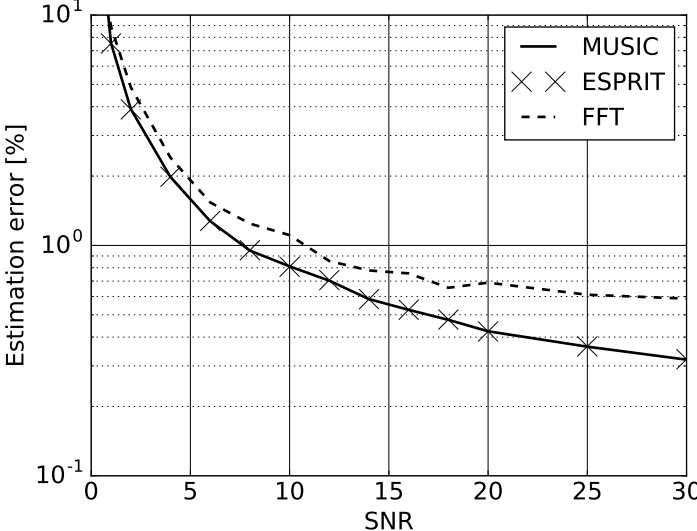

**Figure 7.** Mean amplitude estimation error for different signal-to-noise ratios (SNR). The SNR refers to the signal of one spectral component; the overall signal of the interferogram is the sum of the intensities, all components. The calculations were performed for 700 interferogram samples, each.

and noise subspaces. Liu et al. (2017) investigates the performance of the MUltiple SIgnal Classification (MUSIC, Schmidt, 1986) and Estimating Signal Parameters via Rotational Invariance Techniques (ESPRIT, Roy and Kailath, 1990) methods in comparison to a classical discrete Fourier transformation. To obtain an amplitude estimation error of 1%, the subspace methods require a signal-to-noise ratio of eight, whereas a conventional Fourier-transform requires a 50% higher signal-to-noise ratio (Figure 7). An additional advantage of the subspace methods is the unbiased spatial frequency estimation compared to the discrete frequency values obtained by a Fourier transformation.

# 8 Conclusions

We presented a design for a CubeSat-sized instrument to obtain mesospheric temperatures. A spatial heterodyne spectrometer is used to measure the rotational structure of the $O_2$ A-band, which is complemented by fore- and detector optics. The size of the entire instrument including a straylight baffle is around 3.5 litres and the mass is less than 5 kilograms. The power consumption is about 6 W and the data-rate 50 kByte/image. The instrument can deliver temperatures at a 1–2 K precision for an integration time of about one minute for nightglow and a few seconds for dayglow. A prototype version of this instrument was tested in March 2017 on a sounding rocket by a student team (Deiml et al., 2017). The instrument survived the rocket launch and worked nominally. Unfortunately, it was not possible to record limb spectra with a stable attitude due to a failure of the detumbling mechanism of the rocket. The next step in this project is the advancement of this instrument for an in-orbit



verification on a satellite. The main requirements on a satellite platform are a stable line-of-sight attitude, which should be a few arc minutes for the time of one measurement (a few seconds) (Kaufmann et al., 2017). The control of that angle could be an order of magnitude less precise, since it can be compensated to some degree by an extended vertical field of view of the instrument.

5 *Competing interests.* The authors declare that they have no conflict of interest.

*Acknowledgements.* We acknowledge the consultancy of the optics section at European Space Agency. Part of this work was supported by the Chinese Scholarship Council.





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
