# Peer review of "A highly miniaturized satellite payload based on a spatial"

_Atmospheric Measurement Techniques, 2017_

## Referee Comment (RC1) · Anonymous Referee #1 · 5 Feb 2018

The manuscript entitled: "A highly miniaturized satellite payload based on a spatial heterodyne spectrometer for atmospheric temperature measurements in the mesosphere and lower thermosphere" by Kaufmann et al. describes the design of a miniaturized SHS instrument to measure thermospheric temperature profiles from a CubeSat in low earth orbit, data analysis steps and the expected measurement performance. This manuscript covers a topic that is of significant interest to the community, it is well written, and is well suited for the journal. I recommend the publication of this manuscript after a number of minor comments and one equation error are addressed and cor-

rected.

(1) Abstract: "with a solid angle of 0.65 degrees" The solid angle unit is "steradian", not degrees. Please correct the specification.

(2) Caption of Figure 2: "'Barth' indicates the number of molecules created by the recombination of atomic oxygen." Judging from the axis label and the rest of the description, I assume that 'Barth' indicates the Volume Emission Rate of the airglow component that is resulting from the Barth mechanism, not a number of molecules. Same comment applies to the A-band and B-band description in this caption.

(3) Figure 2: This figure could be improved by (a) omitting the number "2" on the left of the y-axis, (b) adding a "K" after the temperature labels (200, 210) on the top right, and (c) omitting the integral sign and 1 3 0 1 on the top right. It is not clear to me what the latter means.

(4) Page 6, line 2+: "The zero frequency of the fringe pattern is at the Littrow wavelength and the spectral peaks of the neighboring wavelengths are spread or heterodyned around this central wavelength." This sentence is not quite clear to me. What is meant with "spectral peaks"? Instead, it might be worth pointing out that the heterodyning effect results in the fact that high spectral resolution can be obtained because small wavenumber changes result in fringes with discernable, low spatial frequency, which can be observed with available imaging detectors.

(5) Page 6, line 5+: For completeness, the authors might consider adding a reference to the first satellite borne SHS instrument: Englert, C. R., M. H. Stevens, D. E. Siskind, J. M. Harlander, and F. L. Roesler (2010), Spatial Heterodyne Imager for Mesospheric Radicals on STPSat-1, J. Geophys. Res., 115, D20306, doi:10.1029/2010JD014398.

(6) Page 6, line 13: This equation is incorrect. The right side is missing the grating groove density (see Harlander et al., ApJ, 1992, Equation (1))

(7) Page 7, line 1: It might be worth adding that "position x" is in the direction parallel

to the dispersion plane.

(8) Page 7, line 9: It is not quite clear to me why the authors say that the spectral resolution is limited by the detector resolution (pixels per length?). I do agree that the Nyquist theorem limits the bandpass, as the authors state.

(9) Page 7, line 17: For completeness the authors might consider adding to this sentence: ". . . by Hilliard and Shepherd (1966) with a Michelson interferometer, and first introduced for SHS by Roesler and Harlander (1990)." The reference is: "Roesler and Harlander, Spatial heterodyne spectroscopy: interferometric performance at any wavelength without scanning, Proc. SPIE 1318, 1990, doi: 10.1117/12.22119."

(10) Table 1: giving the clear aperture in PI times radius squared is a little confusing. I recommend listing the aperture diameter or radius.

(11) The authors might consider giving the field of view dimensions in both directions, so that a reader can verify that the etendue is the product of the field of view solid angle and the aperture area.

(12) Figure 4: The top axis suggests that the detector is 0.8 cm wide, but using a pixel pitch of 5.04 microns and 840 pixels per row only results in a width of about 0.4 cm. Please check.

(13) Figure 4: The caption states: "Note: I will update the figure later on". Please provide the correct figure (in case this is not the correct one), and please include what local time this simulation was made for, since day and night profiles are significantly different.

(14) Page 9, line 14+: Depending on the field of view orientation with respect to the satellite velocity direction, the scene is scanned through the field of view, which, for a 60 second exposure, can be significant. It might be worth pointing that out.

(15) Page 9, line 18: Please add that the 66mm are the diameter, since some dimensions are given as radius and some as diameter throughout the paper.

(16) Figure 5: The first sentence of the captions claims that the figure includes the filter, the second sentence says that it is not shown. Please clarify.

(17) Page 11, line 11: It might be worth mentioning the Modulation Transfer Function of the detector optics here, since it will potentially influence the modulation of the higher frequency fringes.

(18) Page 11, line 16: The authors state: ". . . 80 photons/s at every spectral point within the interferogram." I think what they mean is ". . . 80 photons/s at every pixel recording the interferogram." The interferogram is in "spatial" space, not in "spectral" space.

(19) Page 12, line 9: The authors state: "Although the dark current at a detector temperature of 20C is a factor of 2.4 larger than the expected atmospheric signal in the nightglow maximum, it does not deteriorate the data processing significantly." This is not clear to me. An additional signal that is 2.4 times larger than the shot noise limited, targeted signal, will increase the noise by a factor of sqrt(3.4), or 80 percent for every interferogram point. I would not call that insignificant.

(20) Page 13, line 14+: It is not clear to me why the first step is required, since all non-uniformities are covered by performing the second step, including the detector non-uniformities.

(21) Page 13, lines 24: Do the authors mean: "Due to the highly compact design of the *front* optics and the use of spherical lenses only. . .", since assessing the effects of the interferometer and detector optics are discussed in the following paragraph?

(22) Page 14, line 9: If I understand this method correctly, it aims to determine the same fringe phases for each rising and falling fringe edge by finding a constant intensity level. This works, if the fringes have a constant offset (non-modulated part), flatfielding has been performed and a correction for modulation efficiency has been performed, prior to finding these edges. If this is what was done here, please include these caveats.

(23) Page 14, line 18+: I suggest reworking the following for clarity from: "total power

in each wave" to "total power for each spectral element".

(24) Page 14, line 20: "known good enough" should be "known well enough"

(25) Page 14, line 21: Using a Hilbert transformation to determine the envelope of the modulated part of the interferogram is fundamentally the same idea as the methods described in Englert et al. 2004 & 2006, where the corresponding complex/imaginary interferogram is generated from the real interferogram. It might be worth pointing that out. In addition, it is not quite clear to me why the authors do not use the fringe phase that can easily be determined using the Hilbert transform to determine the phase distortion. Since it does not rely on the above caveats, it appears to be a more resilient method than using the constant intensity level to find constant phase positions, as described immediately above this section.

(26) Page 14, line 21: The reference Liu et al. 25 (2017) is not yet published as of the submission of this manuscript and could not be accessed by the reviewer. Please, at least, include the final citation.

(27) Section 7: Please mention that this method explicitly requires a-priori information. It would be beneficial if you could comment on whether this is similar to fitting the line strengths of the known lines to the spectrum obtained with an FFT. The FFT does not destroy information, so all additional information has to be from a-priori knowledge.

(28) If the authors have a 3D design image of the instrument design, it would benefit the paper to include it, rather than just stating that there is a design that fits into 3.5 liters. (optional)

(29) From Figure 4, I assume that the instrument will observe the limb between tangent point altitudes of 60km and 120km. Can you please comment on the case in which the airglow extends above 120km during the day? Presumably, the temperature retrieval for at least the highest altitudes will be affected.

(30) Please comment on any thermal effects that are likely to be encountered on orbit,

including thermoelastic distortion of the optics, which can affect the focus of the fringes and therefore the modulation transfer function, depending on fringe frequency (larger effect on high frequency fringes), which might have a significant effect on the relative line strength determination.

(31) It is not clear to me how a 3.5 liter instrument will fit into a 3U CubeSat. Are the authors thinking of further miniaturization?

---

## Referee Comment (RC2) · Anonymous Referee #2 · 19 Feb 2018

This manuscript describes the optical design, radiometric characterization and calibration of a CubeSat-sized limb sounding instrument for measuring temperatures in the Earth's mesosphere and lower thermosphere using emission from the oxygen A-band. The instrument is a field-widened Spatial Heterodyne Spectrometer (SHS) with an effective spectral resolving power of 8000, an altitude resolution of 1.5 km on the limb over a 60 km field of view. My overall impression of the manuscript is positive. It provides sufficient context, a clear description of the work done and analysis of the results. That said there are a few issues that should be addressed to clarify the manuscript.

[Figure]

Specific Comments:

- The diffraction grating groove density (or equivalently, groove spacing) is missing from equation 1.

- The paragraph following equation 4 suggests that "the effective spectral resolution or bandpass is often limited by the detector resolution". Although the bandpass is limited by the detector resolution, as indicated by equation 5, the spectral resolution is not. The spectral resolution is independent of the detector resolution as it depends on two things: 1. The path difference provided by interferometer (as correctly captured in equation 4) and 2. any apodization functions applied to the interferogram.

- The final paragraph of section 3 indicates that the design of the instrument was performed in a collimated configuration. I assume this to mean that only incident rays parallel to the optical axis were used during the optimization of the design. It would be useful to indicate the effect of converging ("focused") beams on the interferogram. Is there any reduction of contrast of the fringes, especially at the edges of the field where the path difference is largest, due to the addition of the off-axis rays? Figure 6 and surrounding discussion suggest so.

- The description of the front optics in section 4 and table 1 indicate that the image of the limb formed on the grating plane is a circle of diameter 7 mm. If the image is circular, the highest and lowest altitude slices at the top and bottom of the image will suffer greatly reduced spectral resolution as they only sample a very small range of the interferometer aperture and only near zero difference. These altitudes will also have significantly reduced etendue due to their small spatial extent. To achieve uniform spectral resolution and etendue for each altitude slice the limb image on the grating and ultimately recorded at the detector should be rectangular as indicated in figure 4. It appears from Figure 5 that there is nothing in the entrance or exit optics that will result in a circular field and is reality limited by the grating or detector, both of which are presumably rectangular.

- The final paragraph in section 4 suggests that a simulation using focused light indicates a reduction of fringe contrast near the edges of the image where the path difference is large. It would be helpful to indicate by how much the contrast is reduced. A plot of intensity vs pixel for a slice through the image shown in figure 6 which shows the fringe reduction would quantify this statement.

- The discussion section 5 of the effect of dark current on the measurement is confusing and on its surface appears to be wrong. It is stated that the dark current at 20 deg C is a factor of 2.4 larger than the maximum atmospheric nighttime signal it does not significantly affect the signal because the multiplex noise is a factor of $5-10$ larger. I don't believe this is the case with the spatially sampled interferogram obtained with SHS. From a noise perspective signal and dark generated electrons are equivalent so if the total number of photons detected in the signal is less than the total dark signal (either on a pixel by pixel or entire detector basis) the noise from the dark signal will dominate. As the authors point out, cooling the detector can reduce the dark signal. It would seem from the discussion that if the dark noise were to be made comparable to the maximum signal, the detector should be cooled to about 10 deg C.

- The discussion in section 6.2 on image and phase distortion correction was confusing. I agree in principle that by measuring the fringe pattern at all wavelengths in the passband of the instrument, corrections for exit optics induced image distortion, which displaces each image point by a fixed distance on the detector, and interferometer induced phase distortion, which changes the phase of a fringe by a fixed amount can be obtained. Note that phase distortion shifts the location of, say, a peak of a fringe by more pixels at low spatial frequency than at high spatial frequency while image distortion would shift a peak by the same number of pixels independent of the frequency of the fringe. That said, it is unclear from the discussion how this will be accomplished in practice. Reference is made to fitting a linear or higher order polynomial correction term to each row, however it is not clear what would be fit: phase?, visibility?, brightness? something else? More discussion here would be helpful.

- Figure 7 and surrounding discussion suggests that an improvement of factor of 2 in noise performance over conventional FFT methods can be achieved by utilizing a-priori information in the fitting process. There isn't enough information in the manuscript to evaluate this technique, however, reference is made to a manuscript in preparation describing the technique and its application to SHS. I look forward to reading this manuscript.

- Both the abstract and conclusions suggest that the instrument can deliver a 1-2 K temperature precision for a one-minute nightglow observation and a few seconds during the day. I would have liked to have seen more support for this statement in the manuscript.

Technical Corrections: - The figure 4 caption indicates that the figure will be updated. Has it?

- There are two missing "C"s to indicate degrees Centigrade in the text immediately following figure 4.

---

## Author Comment (AC1) · 27 Apr 2018

**Reply to the reviewers' comments: Reviewer #1**

General

We thank the reviewer for carefully reading the manuscript and his/ her constructive and helpful comments and suggestions. They helped us to improve the paper in several

aspects. Before we consider them point by point, we like to make the following general remarks:

- The section about the instrument characterization was criticized by the reviewers and we restructured it.

- Further analysis of the spectral power estimation using subspace methods revealed some problems when dealing with interferograms with finite spectral resolution. For the figures shown in the first version of the manscript, infinite spectral resolution was assumed, which is not realistic. To handle finite spectral resultion with subspace methods, the number of spectral components has to be increased, but to develop concepts for further analysis of this kind of data is ongoing work not in the shape to be presented here. Therefore, all analyses and performance assessment in the second version of the manscript is based on conventional Fourier transformations.

- We corrected the value for the etendue of the instrument, which referred to the full circular aperture, but we use an inner rectangular for the later analysis, only. This reduces the orginal value of 0.014 cm$^2$ sr by about 1/3.

- some numbers (like the image size or spectral resolution) differed slightly across the document and were harmonized

Point by point response

*1.: Abstract: "with a solid angle of 0.65 degrees" The solid angle unit is "steradian", not degr*
**Reply**: agreed, we changed solid angle to acceptance angle

**2.**: *Caption of Figure 2: "'Barth' indicates the number of molecules created by the recombination of atomic oxygen." Judging from the axis label and the rest of the description, I assume that 'Barth' indicates the Volume Emission Rate of the airglow component that is resulting from the Barth mechanism, not a number of molecules. Same comment applies to the A-band and B-band description in this caption.*

**Reply**: agreed and corrected.

**3.**: *Figure 2: This figure could be improved by (a) omitting the number "2" on the left of the y-axis, (b) adding a "K" after the temperature labels (200, 210) on the top right, and (c) omitting the integral sign and 1 3 0 1 on the top right. It is not clear to me what the latter means.*

**Reply**: agreed, this was a pdf problem in the final document, corrected

**4.**: *Page 6, line 2+: "The zero frequency of the fringe pattern is at the Littrow wavelength and the spectral peaks of the neighboring wavelengths are spread or heterodyned around this central wavelength." This sentence is not quite clear to me. What is meant with "spectral peaks"? Instead, it might be worth pointing out that the heterodyning effect results in the fact that high spectral resolution can be obtained because small wavenumber changes result in fringes with discernable spatial frequency, which can be observed with available imaging detectors.*

**Reply**: agreed, we changed the wording as suggested.

**5.**: *Page 6, line 5+: For completeness, the authors might consider adding a reference to the first satellite borne SHS instrument: Englert, C. R., M. H. Stevens, D. E. Siskind, J. M. Harlander, and F. L. Roesler (2010), Spatial Heterodyne Imager for Mesospheric Radicals on STPSat-1, J. Geophys. Res., 115, D20306, doi:10.1029/2010JD014398.*

**Reply**: added

*6.: Page 6, line 13: This equation is incorrect. The right side is missing the grating groove density (see Harlander et al., ApJ, 1992, Equation (1))*

**Reply**: agreed, the equation was corrected.

*7.: Page 7, line 1: It might be worth adding that "position x" is in the direction parallelto the dispersion plane.*

**Reply**: agreed, 'parallel to the dispersion plane' was added.

*8.: Page 7, line 9: It is not quite clear to me why the authors say that the spectral resolution is limited by the detector resolution (pixels per length?). I do agree that the Nyquist theorem limits the bandpass, as the authors state.*

**Reply**: agreed. Our statement was imprecise. We meant, that the choice of the grating groove number in combination with the detector pixel number determines (and limits) the spectral resolution and the bandpass. Not to confuse the reader, we changed the wording to 'The bandpass of an SHS is limited by the detector resolution by the Nyquist theorem'.

*9.: Page 7, line 17: For completeness the authors might consider adding to this sentence: ". . . by Hilliard and Shepherd (1966) with a Michelson interferometer, and first introduced for SHS by Roesler and Harlander (1990)." The reference is: "Roesler and Harlander, Spatial heterodyne spectroscopy: interferometric performance at any wavelength without scanning, Proc. SPIE 1318, 1990, doi: 10.1117/12.22119."*

**Reply**: agreed, reference was added

*10.: Table 1: giving the clear aperture in PI times radius squared is a little confusing. I recommend listing the aperture diameter or radius.*

**Reply**: agreed, diameter is now given

**11.**: *The authors might consider giving the field of view dimensions in both directions, so that a reader can verify that the etendue is the product of the field of view solid angle and the aperture area.*

**Reply**: Since we have a circular aperture, the etendue is simply given by the solid angle of the spherical cap and the aperture area. We prefer to keep the numerical value instead of giving a formula.

**12.**: *Figure 4: The top axis suggests that the detector is 0.8 cm wide, but using a pixel pitch of 5.04 microns and 840 pixels per row only results in a width of about 0.4 cm. Please check.*

**Reply**: agreed, the figure was updated in two aspects: The localization plane scale was corrected and nighttime data is shown.

**13.**: *Figure 4: The caption states: "Note: I will update the figure later on". Please provide the correct figure (in case this is not the correct one), and please include what local time this simulation was made for, since day and night profiles are significantly different.*

**Reply**: agreed, we added 'nighttime' in the figure caption and updated the data shown

**14.**: *Page 9, line 14+: Depending on the field of view orientation with respect to the satellite velocity direction, the scene is scanned through the field of view, which, for a 60 second exposure, can be significant. It might be worth pointing that out.*

**Reply**: agreed, we added 'or smeared out during the exposure of the image'

**15.**: *Page 9, line 18: Please add that the 66mm are the diameter, since some dimensions are given as radius and some as diameter throughout the paper.*

**Reply**: done

*16.: Figure 5: The first sentence of the captions claims that the figure includes the filter, the second sentence says that it is not shown. Please clarify.*

**Reply**: agreed, the figure was updated and shows the filter now

*17.: Page 11, line 11: It might be worth mentioning the Modulation Transfer Function of the detector optics here, since it will potentially influence the modulation of the higher frequency fringes.*

**Reply**: Agreed, see also comment of referee 2 on the temperature dependence of the MTF. From our point of view, the MTF is a good *qualitative* indicator for the optical performance of the system, but it cannot be used to *quantify* the visibility reduction due to aberrations, out-of-focus configuration, etc.. To comment on the temperature dependence of the entire optical setup, we give some remarks on the MTF and not on the interferogram contrast, because the simulation is very time consuming and not available for the publication timeframe of this work. We added the following text in the manuscript: 'The SHS has a fairly well athermal design, but the foci and the modulation transfer function (MTF) of the entire optical system depend more on temperature. For low spatial frequencies, this effect is small, but for the highest spatial frequencies seen by the instrument, the MTF reduces from about 85% at $20^{o}$C to about 70% at $0^{o}$C. Further simulations and comparison with measurements are in preparation.'

*18.: Page 11, line 16: The authors state: ". . . 80 photons/s at every spectral point within the interferogram." I think what they mean is ". . . 80 photons/s at every pixel recording the interferogram." The interferogram is in "spatial" space, not in "spectral" space.*

**Reply**: agreed, corrected

*19.: Page 12, line 9: The authors state: "Although the dark current at a detector temperature of 20C is a factor of 2.4 larger than the expected atmospheric signal in the nightglow maximum, it does not deteriorate the data processing significantly." This is*

[Figure]

*not clear to me. An additional signal that is 2.4 times larger than the shot noise limited, targeted signal, will increase the noise by a factor of sqrt(3.4), or 80 percent for every interferogram point. I would not call that insignificant.*

**Reply**: agreed, our wording was not correct and referred to the signal of a single emission line. We give the expected photon flux per pixel earlier in the chapter (40 ph/s, not given explicitly in the previous version) and changed the statement related to the significance of the dark current in the following way: 'At 20°C, the dark current is at least a factor of 5 lower than the atmospheric signal in the emission layer maximum and therefore not a dominant source of random noise at these altitudes. This becomes more critical at other altitudes and for higher detector temperatures.'

*20.: Page 13, line 14+: It is not clear to me why the first step is required, since all non-uniformities are covered by performing the second step, including the detector non-uniformities.*

**Reply**: agreed, we removed the first step from the text. Nevertheless we performed this step, mainly to select a 'good' detector from a batch of detectors.

*21.: Page 13, lines 24: Do the authors mean: "Due to the highly compact design of the \*front\* optics and the use of spherical lenses only. . .", since assessing the effects of the interferometer and detector optics are discussed in the following paragraph?*

**Reply**: We expect to see image distortions mainly from the detector optics, although we also want to quantify distortions introduced by the entire system, which requires a test image to be positioned in front of the front optics. The interferograms can be used to quantify image distortions of the camera optics, but this gives information in the interferogram-dimension, only. The corresponding distortions are likely the same in the other dimension as well, but the test image is a good way to verify this assumption. We re-ordered the entire chapter on 'Instrument Characterization' and hope that this point becomes clearer now.

**22.***: Page 14, line 9: If I understand this method correctly, it aims to determine the same fringe phases for each rising and falling fringe edge by finding a constant intensity level. This works, if the fringes have a constant offset (non-modulated part), flatfielding has been performed and a correction for modulation efficiency has been performed, prior to finding these edges. If this is what was done here, please include these caveats.*

**Reply**: We will use an adaptive edge detection algorithm, which will circumvent the points you mentioned. This was misleading in the first version of this manucscript and we added 'adaptive edge detection' to make this point clear. We also like to mention in this context, that we changed the experimental setup to perform these measurements from an integrating sphere to a homogeneized laser beam. The reason for this modification is the difficulty to project the light of the sphere (with its curved walls) into infinity. We therefore will use microlens arrays to homogeneize the laser light. We changed the text in the following way: To characterize and quantify the modulated part of the intensity, an optical setup with a tunable laser is used. First, the laser light is homogeneized using microlens arrays and imaged onto a rotating diffusor. The laser spot on the diffusor is set to infinity by a large lens, such that the full aperture of the instrument is uniformly illuminated by plane waves with a divergence of at least $\pm 0.65^o$. The laser frequency and power are continuously monitored during the measurement. The laser power and the flux are calibrated before the measurements are taken.'

**23.***: Page 14, line 18+: I suggest reworking the following for clarity from: "total power in each wave" to "total power for each spectral element".*

**Reply**: agreed, changed accordingly

**24.***: Page 14, line 20: "known good enough" should be "known well enough"*

**Reply**: changed

**25.***: Page 14, line 21: Using a Hilbert transformation to determine the envelope of the modulated part of the interferogram is fundamentally the same idea as the methods*

*described in Englert et al. 2004 & 2006, where the corresponding complex/imaginary interferogram is generated from the real interferogram. It might be worth pointing that out. In addition, it is not quite clear to me why the authors do not use the fringe phase that can easily be determined using the Hilbert transform to determine the phase distortion. Since it does not rely on the above caveats, it appears to be a more resilient method than using the constant intensity level to find constant phase positions, as described immediately above this section.*

**Reply**: we added the reference and agree on this comment. The interference pattern is also used to verify and to correct the image distortion orthogonal to the interferogram direction in-orbit, if needed.

*26.: Page 14, line 21: The reference Liu et al. 25 (2017) is not yet published as of the submission of this manuscript and could not be accessed by the reviewer. Please, at least, include the final citation.*

**Reply**: This manuscript is still in preparation. Due to some problems with the interpretation of that data, we omitted this part

*27.: Section 7: Please mention that this method explicitly requires a-priori information. It would be beneficial if you could comment on whether this is similar to fitting the line strengths of the known lines to the spectrum obtained with an FFT. The FFT does not destroy information, so all additional information has to be from a-priori knowledge.*

**Reply**: see above, we removed this part

*28.: If the authors have a 3D design image of the instrument design, it would benefit the paper to include it, rather than just stating that there is a design that fits into 3.5 liters. (optional)*

**Reply**: A design image is now included in the conclusion section

**29.**: *From Figure 4, I assume that the instrument will observe the limb between tangent point altitudes of 60km and 120km. Can you please comment on the case in which the airglow extends above 120km during the day? Presumably, the temperature retrieval for at least the highest altitudes will be affected.*

**Reply**: This is true and a general retrieval 'problem'. In existing retrievals, the regularization parameters of a constrained retrieval setup are chosen in such a way, that the information obtained from the upper most measurement altitude(s) is spread over a broad altitude regime, resulting in a very broad vertical resolution of the retrieved quantities. The corresponding temperatures are not very useful for further analyses, but a smooth transition into some a priori data is assured by this method. We feel that this information is difficult to place in this manuscript and prefer to give a more detailed discussion on the retrieval in a separate manuscript

**30.**: *Please comment on any thermal effects that are likely to be encountered on orbit, including thermoelastic distortion of the optics, which can affect the focus of the fringes and therefore the modulation transfer function, depending on fringe frequency (larger effect on high frequency fringes), which might have a significant effect on the relative line strength determination.*

**Reply**: Agreed. We calculated the MTF and added the following text: 'The SHS has a fairly well athermal design, but the foci and the modulation transfer function (MTF) of the entire optical system depend more on temperature. For low spatial frequencies, this effect is small, but for the highest spatial frequencies seen by the instrument, the MTF reduces from about 85% at $20^o$C to about 70% at $0^o$C. Further simulations and comparison with measurements are in preparation.'

**31.**: *It is not clear to me how a 3.5 liter instrument will fit into a 3U CubeSat. Are the authors thinking of further miniaturization?*

**Reply**: The optical instrument itself fits into about 1.5 litres. Deiml et al. [2014] made

a concept study of an extendable baffle to fit the entire instrument into a 3-unit Cube-Sat. We added the following text in the manuscript: 'The utilization of an extendable baffle and some minor design modifications allows to fly the instrument on a three–unit CubeSat.' One design modifications is to decrease the length of the detector optics by a few millimetres by using some aspheres instead of spherical lenses. We made a corresponding design, but it was not persued for budget reasons. For a CubeSat mission we favour a 6-unit spacecraft to relax the compactness of the entire instrument, to avoid the risk of an extendable baffle, to allow for more power, and a few other reasons. M. Deiml, M. Kaufmann, P. Knieling, F. Olschewski, P. Toumpas, M. Langer, M. Ern, R. Koppmann, and M. Riese, "Dissect: development of a small satellite for climate research," Proceedings of the 65th International Astronautical Congress, Toronto, Canada, no. IAC-14,B5,1,10,x22911, 2014.

---

## Author Comment (AC2) · 27 Apr 2018

**Reply to the reviewers' comments: Reviewer #2**

General

We thank the reviewer for carefully reading the manuscript and his/ her constructive and helpful comments and suggestions. They helped us to improve the paper in several

aspects. Before we consider them point by point, we like to make the following general remarks:

- The section about the instrument characterization was criticized by the reviewers and we restructured it.

- Further analysis of the spectral power estimation using subspace methods revealed some problems when dealing with interferograms with finite spectral resolution. For the figures shown in the first version of the manscript, infinite spectral resolution was assumed, which is not realistic. To handle finite spectral resultion with subspace methods, the number of spectral components has to be increased, but to develop concepts for further analysis of this kind of data is ongoing work not in the shape to be presented here. Therefore, all analyses and performance assessment in the second version of the manscript is based on conventional Fourier transformations.

- We corrected the value for the etendue of the instrument, which referred to the full circular aperture, but we use an inner rectangular for the later analysis, only. This reduces the orginal value of 0.014 cm$^2$ sr by about 1/3.

- some numbers (like the image size or spectral resolution) differed slightly across the document and were harmonized

Point by point response

**1.:** *The paragraph following equation 4 suggests that "the effective spectral resolution or bandpass is often limited by the detector resolution". Although the bandpass is limited by the detector resolution, as indicated by equation 5, the spectral resolution is not. The spectral resolution is independent of the detector resolution as it depends on*
*two things: 1. The path difference provided by interferometer (as correctly captured in equation 4) and 2. any apodization functions applied to the interferogram.*

**Reply**: agreed, we changed the text accordingly.

*2.: The final paragraph of section 3 indicates that the design of the instrument was performed in a collimated configuration. I assume this to mean that only incident rays parallel to the optical axis were used during the optimization of the design. It would be useful to indicate the effect of converging ("focused") beams on the interferogram. Is there any reduction of contrast of the fringes, especially at the edges of the field where the path difference is largesft, due to the addition of the off-axis rays? Figure 6 and surrounding discussion suggest so.*

**Reply**: This text was misleading and we removed it. We designed and optimized the SHS using converging beams, but we were not able to calculate interferograms in this configuration at the time of the SHS design (not supported by raytracing software). This worked only for collimated light and therefore our first interferograms were calculated for collimated light only. However, in the meantime we control the raytracing software in such a way that we are able to calculate interferograms for the focused configuration as well, which we show later in the paper.

*3.: The description of the front optics in section 4 and table 1 indicate that the image of the limb formed on the grating plane is a circle of diameter 7 mm. If the image is circular, the highest and lowest altitude slices at the top and bottom of the image will suffer greatly reduced spectral resolution as they only sample a very small range of the interferometer aperture and only near zero difference. These altitudes will also have significantly reduced etendue due to their small spatial extent. To achieve uniform spectral resolution and etendue for each altitude slice the limb image on the grating and ultimately recorded at the detector should be rectangular as indicated in figure 4. It appears from Figure 5 that there is nothing in the entrance or exit optics that will result in a circular field and is reality limited by the grating or detector, both of which*
*are presumably rectangular.*

**Reply**: We agree, our wording was imprecise and not correct in all points. We changed the text in the following way: 'The front-optics consists of four lenses, which image an object at infinity onto a square with an edge length of 7 mm on the virtual image of the gratings. This corresponds to a theoretical spectral resolution of about 16,800. The maximum chief ray angle extent is about $1.9^o$, such that a rectangular object with an angular extent of $1.3^o$ can be captured without vignetting.'

*4.: - The final paragraph in section 4 suggests that a simulation using focused light indicates a reduction of fringe contrast near the edges of the image where the path difference is large. It would be helpful to indicate by how much the contrast is reduced. A plot of intensity vs pixel for a slice through the image shown in figure 6 which shows the fringe reduction would quantify this statement.*

**Reply**: Agreed, we added a 1d plot of the interferogram and the following text: 'The detection plane was placed between the focal planes for the on axis and the $0.65^o$ off axis light source points as a compromise, and closer to the latter one to enhance the visibility on the edges of the interferogram. Nevertheless, the visibility reduction is about 1/3 towards the edges. Interestingly, the highest visibility is achieved by placing the detector plane outside both focal planes in a plane which is near the on axis focal point. The suspected reason is that the shape of the focal spots, which are blurred by aberrations resulting in a reduction of visibility, becomes more compact if the detector plane is positioned slightly out of the on axis focus, yielding to higher contrast (Figure 7).'

*5.: The discussion section 5 of the effect of dark current on the measurement is confusing and on its surface appears to be wrong. It is stated that the dark current at 20 deg C is a factor of 2.4 larger than the maximum atmospheric nighttime signal it does not significantly affect the signal because the multiplex noise is a factor of 5 – 10 larger. I don't believe this is the case with the spatially sampled interferogram obtained with*

*SHS. From a noise perspective signal and dark generated electrons are equivalent so if the total number of photons detected in the signal is less than the total dark signal (either on a pixel by pixel or entire detector basis) the noise from the dark signal will dominate. As the authors point out, cooling the detector can reduce the dark signal. It would seem from the discussion that if the dark noise were to be made comparable to the maximum signal, the detector should be cooled to about 10 deg C.*

**Reply**: agreed, our wording was not correct and referred to the signal of a single emission line. We give the expected photon flux per pixel earlier in the chapter (40 ph/s, not given explicitly in the previous version) and changed the statement related to the significance of the dark current in the following way: 'At 20°C, the dark current is at least a factor of 5 lower than the atmospheric signal in the emission layer maximum and therefore not a dominant source of random noise at these altitudes. This becomes more critical at other altitudes and for higher detector temperatures.'

*6.: The discussion in section 6.2 on image and phase distortion correction was confusing. I agree in principle that by measuring the fringe pattern at all wavelengths in the passband of the instrument, corrections for exit optics induced image distortion, which displaces each image point by a fixed distance on the detector, and interferometer induced phase distortion, which changes the phase of a fringe by a fixed amount can be obtained. Note that phase distortion shifts the location of, say, a peak of a fringe by more pixels at low spatial frequency than at high spatial frequency while image distortion would shift a peak by the same number of pixels independent of the frequency of the fringe. That said, it is unclear from the discussion how this will be accomplished in practice. Reference is made to fitting a linear or higher order polynomial correction term to each row, however it is not clear what would be fit: phase?, visibility?, brightness? something else? More discussion here would be helpful.*

**Reply**: agreed, we have re-written the entire chapter.

*7.: Figure 7 and surrounding discussion suggests that an improvement of factor of 2 in*

*noise performance over conventional FFT methods can be achieved by utilizing a-priori information in the fitting process. There isn't enough information in the manuscript to evaluate this technique, however, reference is made to a manuscript in preparation describing the technique and its application to SHS. I look forward to reading this manuscript.*

**Reply**: section removed, see general remark.

**8.:** *Both the abstract and conclusions suggest that the instrument can deliver a 1-2 K temperature precision for a one-minute nightglow observation and a few seconds during the day. I would have liked to have seen more support for this statement in the manuscript. Has it?*

**Reply**: We added the following text: 'The required signal-to-noise ratio to achieve a given temperature precison was determined by Monte-Carlo simulations: First, a simulated spectrum with the optical resolving power of 16,800 was calculated. This spectrum was inverse Fourier-transformed and white noise was added. In the next step, the spectral power in the various frequencies was estimated by applying a Fourier-transformation using a windowing function. The resulting spectra were then used to retrieve an atmospheric temperature profile and some other instrumential parameters, such as the spectral resolution of the data. Considering the intensity of the A-band signal of the nightglow layer maximum and the detector performance, the expected signal-to-noise ratio for a vertical resolution of 1.5 km and an integration time of 60 s will be 10-20 in the nightglow maximum, resulting in a retrieved temperature precision of 1–2 K.'

**9.:** *Technical Corrections: - The figure 4 caption indicates that the figure will be updated. There are two missing "C"s to indicate degrees Centigrade in the text immediately following figure 4.*

**Reply**: agreed, this was a pdf problem in the final document, corrected